# AA-PINN: ATTENTION AUGMENTED PHYSICS IN-FORMED NEURAL NETWORKS

## ABSTRACT

Physics Informed Neural Networks has been quite successful in modelling the complex nature of fluid flow. Computational Fluid Dynamics using parallel processing algorithms on GPUs have considerably reduced the time to solve the Navier Stokes Equations. CFD based approaches uses approximates to make the modelling easy but it comes at the cost of decrease in accuracy. In this paper, we propose an attention based network architecture named AA-PINN to model PDEs behind fluid flow. We use a combination of channel and spatial attention module. We propose a novel loss function which is more robust in handling the initial as well as boundary conditions imposed. Using evaluation metrics like RMSE, divergence and thermal kinetic energy, our network outperforms previous PINNs for modelling Navier Stokes and Burgers Equation.

## 1 INTRODUCTION

Computational Fluid Dynamics (CFD) has become the core technology behind almost every fluid simulation. Fluid mechanics has been traditionally concerned with big data, thus making deep learning an obvious choice in modelling the inherent complexity of the problem. Neural Networks of late has been quite successful in understanding, predicting, optimizing, and controlling fluid flows. Neural Network has proven to improve optimization performance and reduce convergence time drastically. Neural network is also used for turbulence modelling and identifying low and high dimensional flow regimes. Deep learning algorithms are able to take into account inherent complexity of the problem thus optimizing for the performance, robustness or convergence for complex tasks. Understanding the physics behind fluid flows is a complex problem which can be solved using neural networks by feeding lots of training data. It helps in providing a general purpose framework for interpretability, generalizability and explainability of the results achieved.

## 2 RELATED WORK

Neural network to solve Reynolds Averaged Navier Stokes Equation was proposed by (Ling et al., 2016). The Reynolds stress term was modelled using DNS equation by (Wang et al., 2017). Neural network was used to model turbulent flows using Large Eddy simulation (Zhou et al., 2019). A convolutional neural network was used to model the velocity field over a cylinder (Jin et al., 2018). (Wu et al., 2020) proposed a similar CNN based method to model the unsteady flow in arbitrary fluid regimes. A thorough study of data driven methods using machine learning approaches for modelling the turbulence was studied by (Duraisamy et al., 2019). (Brunton et al., 2020) also did a comprehensive study of machine learning approaches for modelling different kind of problems in fluid mechanics. (Raissi et al., 2017) proposed physics informed neural networks for solving nonlinear partial differential equations using neural network.

This work was further improved in (Raissi et al., 2019). The theoretical exact solution of the 3d Navier Stokes equation was shown by (Ethier and Steinman, 1994). Deep feedforward neural networks was used (Lui and Wolf, 2019) for modelling complex flow regimes. CNN were used for making faster fluid simulation (Tompson et al., 2017). A novel neural network was proposed for solving the function approximation and inverse PDE problems (Meng and Karniadakis, 2020). (Khoo et al., 2021) used neural network for solving parametric PDEs. (Meng et al., 2020) proposed a neural network for solving unsteady PDEs. Bayesian neural network was used to quantify the uncertainty

while solving PDEs (Yang et al., 2021). Data driven approaches for solving PDEs was proposed by (Long et al., 2018) and (Long et al., 2019). c(Sirignano et al., 2020) used neural network for solving PDEs in the context of large-eddy simulation.

(Bar and Sochen, 2019) was one of the first works to use unsupervised learning for solving PDEs. (Thuerey et al., 2020) proposed a deep learning approach for solving Reynolds-averaged Navier-Stokes equation around airfoils. Another approach for solving PDEs using deep learning was used (Miyanawala and Jaiman, 2017) in the context of unsteady wake flow dynamics. A comprehensive study of deep learning approaches for modelling and solving fluid mechanics problem was done by (Kutz, 2017). (Ranade et al., 2021) proposed a deep learning based solver for Navier–Stokes equations using finite volume discretization. Neural networks was used for solving incompressible Navier-Stokes equations (Jin et al., 2021). A method for predicting turbulent flows using deep learning was proposed by (Wang et al., 2020).

Our main contributions can be summarized as:

• A novel network architecture combining channel and spatial attention mechanism is used for modelling the inherent complexity in fluid flow problems.

• We train and test our network using a more robust loss function for solve PDEs behind incompressible Navier Stokes and Burgers Equation.

• Our network achieves better results than previous PINNs using commonly used evaluation metrics while still running at good enough speed.

## 3 BACKGROUND

### 3.1 NAVIER STOKES EQUATION

The incompressible transient two dimensional Navier-Stokes equations for mass and momentum conservation are written as defined in the below set of equations:

$$\nabla \cdot \mathbf{u} = 0 \tag{1}$$

$$u_x \frac{\partial u_x}{\partial x} + u_y \frac{\partial u_x}{\partial y} = -\frac{1}{\rho}\frac{\partial p}{\partial x} + \nu\nabla^2 u_x + g_x \tag{2}$$

$$u_x \frac{\partial u_y}{\partial x} + u_y \frac{\partial u_y}{\partial y} = -\frac{1}{\rho}\frac{\partial p}{\partial y} + \nu\nabla^2 u_y + g_y \tag{3}$$

in which $u$ is the velocity field (with $x$ and $y$ components for 2 dimensional flows). Here $g$ represents the gravitational acceleration and $\mu$ the dynamic viscosity of the fluid.

### 3.2 MOMENTUM EQUATIONS

When the difference operators are expanded using uniform grid spacing $h$ and time step $k$ results in:

$$
\begin{aligned}
u_{i,j} - \frac{k\nu}{h^2}\left(u_{i-1,j} + u_{i,j-1} - 4u_{i,j} + u_{i,j+1} + u_{i+1,j}\right) + \frac{k}{h}\left(\bar{u}_{i+1,j}^n\bar{u}_{i+1,j} - \bar{u}_{i,j}^n\bar{u}_{i,j}\right)\\
+ \frac{k}{h}\left(\bar{v}_{i,j}^n\widetilde{u}_{i,j} - \bar{v}_{i,j-1}^n\widetilde{u}_{i,j-1}\right) = u_{i,j}^n - \frac{k}{h}\left(p_{i+1,j} - p_{i,j}\right)
\end{aligned}
\tag{4}
$$

Where variables without superscripts denote advanced time level results to be computed. Using the formulas of the averages and collecting the terms results in equation below:

$$- A_1 u_{i-1,j} - A_2 u_{i,j-1} + A_3 u_{i,j} - A_4 u_{i,j+1} - A_5 u_{i+1,j} = b_{i,j} - \frac{k}{h}\left(p_{i+1,j} - p_{i,j}\right) \tag{5}$$

The various coefficients in the above equation are given using the set of equations as follows:

$$A_1 = \frac{k}{h}\left(\frac{\nu}{h} + \frac{1}{2}\bar{u}^n_{i,j}\right) \tag{6}$$

$$A_2 = \frac{k}{h}\left(\frac{\nu}{h} + \frac{1}{2}\bar{v}^n_{i,j-1}\right) \tag{7}$$

$$A_3 = 1 + 4\frac{k\nu}{h^2} + \frac{k}{2h}\left(\bar{u}^n_{i+1,j} - \bar{u}^n_{i,j} + \bar{v}^n_{i,j} - \bar{v}^n_{i,j-1}\right) \tag{8}$$

$$A_4 = \frac{k}{h}\left(\frac{\nu}{h} - \frac{1}{2}\bar{v}^n_{i,j}\right) \tag{9}$$

$$A_5 = \frac{k}{h}\left(\frac{\nu}{h} - \frac{1}{2}\bar{u}^n_{i+1,j}\right) \tag{10}$$

It is to be noted that in the continuous equations, we ignored effects of correction quantities in advective and diffusive terms. The u-component velocity correction can be written in the form as defined in Equation below:

$$u'_{i,j} = \frac{k}{A_3 h}\left(p'_{i,j} - p'_{i+1,j}\right) \tag{11}$$

We now present analogous results for the y-momentum equation. The v-component velocity correction can be written in the form as defined in Equation below:

$$v'_{i,j} = \frac{k}{B_3 h}\left(p'_{i,j} - p'_{i,j+1}\right) \tag{12}$$

## 3.3  PRESSURE POISSON EQUATION

By substituting the velocity corrections into the discrete continuity equation for grid cell $(i, j)$ results in:

$$\frac{u_{i,j} - u_{i-1,j}}{h_x} + \frac{v_{i,j} - v_{i,j-1}}{h_y} = 0 \tag{13}$$

After substituting the decomposed velocity components, the above equation can be written as:

$$\frac{(u^* + u')_{i,j} - (u^* + u')_{i-1,j}}{h_x} + \frac{(v^* + v')_{i,j} - (v^* + v')_{i,j-1}}{h_y} = 0 \tag{14}$$

For simplicitiy, we set $h_x = h_y = $ h, and rewrite this as:

$$
\begin{aligned}
-\frac{1}{A_{3,i-1,j}}p'_{i-1,j} - \frac{1}{B_{3,i,j-1}}p'_{i,j-1} + \left(\frac{1}{A_{3,i,j}} + \frac{1}{A_{3,i-1,j}} + \frac{1}{B_{3,i,j}} + \frac{1}{B_{3,i,j-1}}\right)p'_{i,j} \\
-\frac{1}{B_{3,i,j}}p'_{i,j+1} - \frac{1}{A_{3,i,j}}p'_{i+1,j} = -\frac{h^2}{k}D^*_{i,j}
\end{aligned}
\tag{15}
$$

It can alternatively written in a more compact form similar to that used for the momentum equations:

$$C_1 p'_{i-1,j} + C_2 p'_{i,j-1} + C_3 p'_{i,j} + C_4 p'_{i,j+1} + C_5 p'_{i+1,j} = d^*_{i,j} \tag{16}$$

The various coefficients in the above equation is defined as follows:

$$C_1 \equiv \frac{1}{A_{3,i-1,j}}, \quad C_2 \equiv \frac{1}{B_{3,i,j-1}}, \quad C_3 \equiv -\left(\frac{1}{A_{3,i,j}} + \frac{1}{A_{3,i-1,j}} + \frac{1}{B_{3,i,j}} + \frac{1}{B_{3,i,j-1}}\right)$$
$$C_4 \equiv \frac{1}{B_{3,i,j}}, \quad C_5 \equiv \frac{1}{A_{3,i,j}}, \quad d_{i,j}^* \equiv \frac{h^2}{k}D_{i,j}^* \tag{17}$$

### 3.4 BURGER'S EQUATION

In one space dimension, the Burger's equation along with Dirichlet boundary conditions is defined using the below set of equations:

$$\begin{aligned} u_t + uu_x - (0.01/\pi)u_{xx} &= 0, \quad x \in [-1,1], \quad t \in [0,1] \\ u(0,x) &= -\sin(\pi x) \\ u(t,-1) &= u(t,1) = 0 \end{aligned} \tag{18}$$

Here, $t_u^i, x_u^i, u_i \ N_u$ i=1 denotes the initial and boundary training data on $u(t,x)$ and $t_f^i, x_f^i \ N_f$ i=1 denotes the collocations points for $f(t,x)$. The loss $MSE_u$ corresponds to the initial and boundary data while $MSE_f$ enforces the structure used by equation at a finite set of collocation points.

## 4 METHOD

### 4.1 SPATIAL ATTENTION MODULE

The spatial attention module is used for capturing the spatial dependencies of the feature maps. The spatial attention (SA) module used in our network is defined below:

$$f_{SA}(x) = f_{sigmoid}\left(W_2\left(f_{ReLU}\left(W_1(x)\right)\right)\right) \tag{19}$$

where $W_1$ and $W_2$ denotes the first and second $1 \times 1$ convolution layer respectively, $x$ denotes the input data, $f_{Sigmoid}$ denotes the sigmoid function, $f_{ReLU}$ denotes the ReLu activation function.

The spatial attention module used in this work is shown in Figure 1:

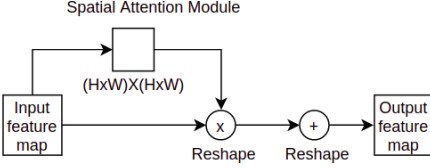

Figure 1: Details of our spatial attention module
.

### 4.2 CHANNEL ATTENTION MODULE

The channel attention module is used for extracting high level multi-scale semantic information. The channel attention (CA) module used in our network is defined below:

$$f_{CA}(x) = f_{sigmoid}(W_2(f_{ReLU}(W_1 f_{AvgPool}^1(x)))) \tag{20}$$

where $W_1$ and $W_2$ denotes the first and second $1 \times 1$ convolution layer, $x$ denotes the input data. $f_{AvgPool}^1$ denotes the global average pooling function, $f_{Sigmoid}$ denotes the Sigmoid function, $f_{ReLU}$ denotes ReLU activation function.

The channel attention module used in this work is shown in Figure 2:

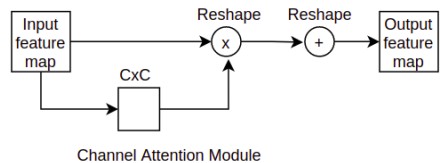

Figure 2: Details of our channel attention module

.

### 4.3 NETWORK ARCHITECTURE

We use deep convolutional neural network in this work. The input is the spatial and temporal co-ordinates of the points in the fluid flow domain. This information is propagated to three Residual blocks sequentially. In between the blocks, channel attention module is used to weight the usefulness of important features and spatial attention module is used for modelling the inter-spatial relationship of features. Fusion operator is used to merge the individual features. The output is the spatio-temporal pressure and velocity fields predicted. The complete network architecture used in this work is shown in Figure 3:

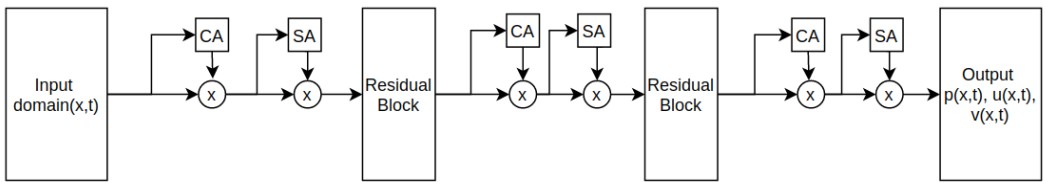

Figure 3: Illustration of our network architecture. A residual block denotes convolution, max pooling, relu activation function and batch normalization layer sequentially; CA and SA denotes channel and spatial attention module respectively; $x$ denotes fusion operator.

### 4.4 LOSS FUNCTIONS

The MSE loss function is used for both the X and Y components of momentum equation which is defined as:

$$MSE_u = \frac{1}{N_u} \sum_{i=1}^{N_u} \left| u\left(t_u^i, x_u^i\right) - u^i \right|^2 \tag{21}$$

$$MSE_f = \frac{1}{N_f} \sum_{i=1}^{N_f} \left| f\left(t_f^i, x_f^i\right) \right|^2 \tag{22}$$

The shared parameters between the neural networks $u(t, x)$ and $f(t, x)$ can be learned by minimizing the mean squared error loss as defined using in the equation below:

$$MSE_t = \alpha MSE_u + \beta MSE_f \tag{23}$$

The weighting coefficients $\alpha$ and $\beta$ are used to balance different terms of the loss function and accelerate convergence in the training process. The individual loss function terms $L_e$, $L_b$ and $L_i$ represent loss function components corresponding to the residual of the Navier-Stokes equations, the boundary conditions, and the initial conditions, respectively. The loss function is defined using the set of equations below:

$$L_e = \frac{1}{N_e} \sum_{i=1}^{4} \sum_{n=1}^{N_e} |e_{VPi}^n|^2 \tag{24}$$

$$L_b = \frac{1}{N_b} \sum_{n=1}^{N_b} |\mathbf{u}^n - \mathbf{u}_b^n|^2 \tag{25}$$

$$L_i = \frac{1}{N_i} \sum_{n=1}^{N_i} |\mathbf{u}^n - \mathbf{u}_i^n|^2 \tag{26}$$

Where $N_b$, $N_i$ and $N_e$ denote the number of training data for different terms. The above 3 terms can be combined to give:

$$L_t = \gamma L_e + \delta L_b + \rho L_i \tag{27}$$

The weighting coefficients $\gamma$, $\delta$ and $\rho$ are used to balance different terms of the loss function and accelerate convergence in the training process. The complete loss function for training the parameters of our network is defined as follows:

$$L_{final} = MSE_t + L_t \tag{28}$$

## 4.5 OPTIMIZATION DETAILS

For a general gradient descent algorithm, the iterative formulation of the parameters of our network can be expressed as:

$$\theta^{(k+1)} = \theta^{(k)} - \eta\gamma\nabla_\theta L_e - \eta\delta\nabla_\theta L_b - \eta\rho\nabla_\theta L_i - \eta\alpha\nabla_\theta MSE_u - \eta\beta\nabla_\theta MSE_f \tag{29}$$

where $\theta$ denotes the parameters of the neural network, namely the weights of all the layers, $k$ is the iteration step, and $\eta$ is the learning rate.

## 4.6 EVALUATION METRICS

Root Mean Square Error (RMSE) is the most popularly used metric for quantifying the prediction performance. The downside of using it is that it only measures indivdual pixel differences. There is a need to check whether the predictions are physically meaningful and preserve desired physical quantities, such as Turbulence Kinetic Energy and Divergence. In this work, we use following metrics for evaluation.

**1. Root Mean Square Error:** We calculate the RMSE of all predicted values from the ground truth for each pixel.

**2. Divergence:** We use the average of absolute divergence over all pixels at each prediction step as an evaluation metric.

**3. Turbulence Kinetic Energy:** The turbulence kinetic energy is characterised by measured root mean square velocity fluctuations as defined by:

$$\left( \overline{(u')^2} + \overline{(v')^2} \right)/2, \quad \overline{(u')^2} = \frac{1}{T} \sum_{t=0}^{T} (u(t) - \bar{u})^2 \tag{30}$$

where $t$ is the time step. We calculate the turbulence kinetic energy for each predicted sample of 100 velocity fields.

### 4.7 IMPLEMENTATION DETAILS

An adaptive optimization algorithm, Adam (Kingma and Ba, 2014), is used to optimize the loss function. The parameters of the neural networks are randomly initialized using the Xavier intitalization scheme. We simulate turbulent channel flow at $Re_\tau = 9.99 \times 10^2$ using our network. The time step value of 0.005 is used for evaluating the residuals our network. We feed the training data using mini-batches to train our network in this study. There are three parts in the input data corresponding to the initial conditions, the boundary conditions and the residuals of equations respectively. We place 100,000 points inside the domain and 25,000 points on the boundary sampled at each time step, and 150,000 points at the initial time step to determine the loss function. The total number of iterations in one training epoch used is 100. The hyper-parameter values are $\alpha = 100$, $\beta = 100$.

## 5 RESULTS

The performance comparison of our network with previous state of the art is shown in Table 1:

Table 1: Comparion of SOTA networks using the number of parameters, the best number of input frames, the best number of accumulated errors for back-propogation and training time for one epoch.

| Models | TF-net | U-net | GAN | ResNet | ConvLSTM | SST | DHPM | Ours |
|---|---|---|---|---|---|---|---|---|
| number of parameters($10^6$) | 15.9 | 25.0 | 26.1 | 21.2 | 11.8 | 49.9 | 2.12 | **0.53** |
| input length | 25 | 25 | 24 | 26 | 27 | 23 | 23 | **20** |
| accumulated errors | 4 | 6 | 5 | 5 | 4 | 5 | 5 | **2** |
| time for one epoch(min) | **0.39** | 0.57 | 0.73 | 1.68 | 45.6 | 0.95 | 4.591 | 0.72 |

The exact and learned dynamics solution for the Burgers equation using our network is shown in Figure 4:

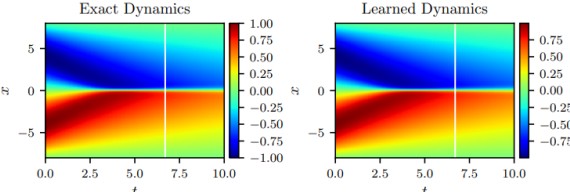

Figure 4: A solution of the Burger's equation (left panel) is compared to the corresponding solution of the learned partial differential equation (right panel).

The exact and learned dynamics solution for the Navier Stokes equation using our network is shown in Figure 5:

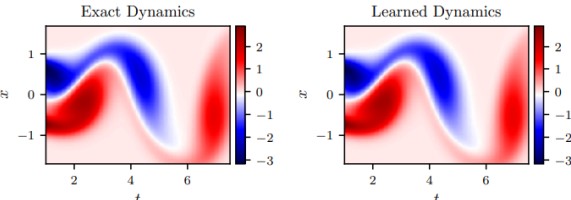

Figure 5: A solution of the NavierStokes equation (left panel) is compared to the corresponding solution of the learned partial differential equation (right panel).

The actual and and the predicted dynamics of the velocity components $u$ and $v$ using our network at different timeframes is shown in Figure 6:

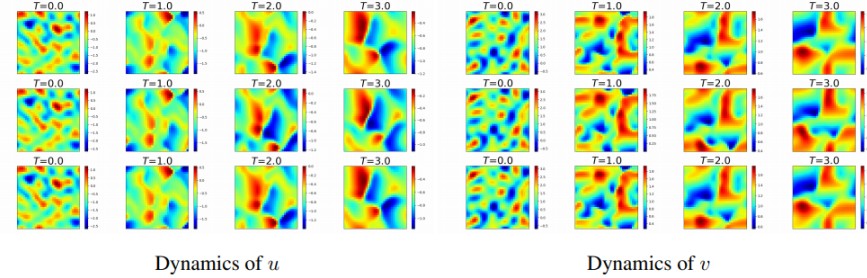

Dynamics of $u$              Dynamics of $v$

Figure 6: The first row shows the images of the true dynamics. The last two rows show the images of the predicted dynamics using our network.

## 5.1 ABLATION STUDIES

A study of with and without using channel and spatial attention module on the performance is shown in Table 2:

Table 2: Ablation study using variations of spatial and channel attention modules.

| Metrics | Only SA | Only CA | Both |
|---|---|---|---|
| number of parameters($10^6$) | 1.23 | 0.71 | **0.53** |
| accumulated errors | 5 | 3 | **2** |
| time for one epoch(min) | 1.05 | 1.16 | **0.72** |

## 6 CONCLUSIONS

In this paper, we present a attention based physics informed neural network named AA-PINN to simulate incompressible Navier Stokes and Burgers Equations. We formulate our network using Pressure-Velocity coupling. The spatial and temporal co-ordinates of the domain are input while instantaneous pressure and velocity fields are output. We use the initial and boundary conditions as supervised data-driven parts, while residual of the Navier-Stokes and Burgers equations as the unsupervised part in the loss function while training our network. We propose a more robust loss function to handle both the boundary conditions as well as initial conditions. We test the performance our network using RMSE, divergence and TKE as the evaluation metrics. We demonstrate our designed network is more robust while modelling the complex flow physics. In the future, we would like to study the effect of attention mechanism for solving compressible and steady Navier Stokes Equations.

## ACKNOWLEDGMENTS

We would like to thank Nvidia for providing the GPUs for this work.

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

## 7 ADDITIONAL MATERIALS

### 7.1 GALERKIN PROCEDURE

Navier-Stokes equations can also be written as using the set of equations below:

$$u_t + \left(u^2\right)_x + (uv)_y = -p_x + \frac{1}{Re}\Delta u \tag{31}$$

$$v_t + (uv)_x + \left(v^2\right)_y = -p_y + \frac{1}{Re}\Delta v \tag{32}$$

$$u_x + v_y = 0 \tag{33}$$

The dependent variables expressed in terms of Fourier Series is defined using the set of equations below:

$$u(x,y,t) = \sum_{\boldsymbol{k}}^{\infty} a_{\boldsymbol{k}}(t)\varphi_{\boldsymbol{k}}(x,y) \tag{34}$$

$$v(x,y,t) = \sum_{\boldsymbol{k}}^{\infty} b_{\boldsymbol{k}}(t)\varphi_{\boldsymbol{k}}(x,y) \tag{35}$$

$$p(x, y, t) = \sum_{\boldsymbol{k}}^{\infty} c_{\boldsymbol{k}}(t) \varphi_{\boldsymbol{k}}(x, y) \tag{36}$$

The exponential are used as basis functions as shown below:

$$\varphi_{\boldsymbol{k}}(x, y) = e^{i\boldsymbol{k}\cdot\boldsymbol{x}} = e^{i(k_1 x + k_2 y)} = e^{ik_1 x} e^{ik_2 y} \tag{37}$$

Substituting the above equation in momentum Navier Stokes Equation results in:

$$i \sum_{\boldsymbol{k}}^{\infty} (k_1 a_{\boldsymbol{k}} + k_2 b_{\boldsymbol{k}}) \varphi_{\boldsymbol{k}} = 0 \tag{38}$$

Since, this is true for all the points in the domain considered, hence it can be written as:

$$k_1 a_{\boldsymbol{k}} + k_2 b_{\boldsymbol{k}} = 0 \quad \forall \boldsymbol{k} \tag{39}$$

On substituting these expansions into the x-momentum equation, we obtain:

$$\frac{\partial}{\partial t} \sum_{\ell} a_\ell \varphi_\ell + \frac{\partial}{\partial x} \sum_{\ell,\boldsymbol{m}} a_\ell a_{\boldsymbol{m}} \varphi_\ell \varphi_{\boldsymbol{m}} + \frac{\partial}{\partial y} \sum_{\ell,\boldsymbol{m}} a_\ell b_{\boldsymbol{m}} \varphi_\ell \varphi_{\boldsymbol{m}} = \\ -\frac{\partial}{\partial x} \sum_{\ell} c_\ell \varphi_\ell + \frac{1}{Re} \left[ \frac{\partial^2}{\partial x^2} \sum_{\ell} a_\ell \varphi_\ell + \frac{\partial^2}{\partial y^2} \sum_{\ell} a_\ell \varphi_\ell \right] \tag{40}$$

The process of commuting, summation and differentiation gives:

$$\sum_{\ell} \dot{a}_\ell \varphi_\ell + i \sum_{\ell,m} (\ell_1 + m_1) a_\ell a_{\boldsymbol{m}} \varphi_\ell \varphi_{\boldsymbol{m}} + i \sum_{\ell,\boldsymbol{m}} (\ell_2 + m_2) a_\ell b_{\boldsymbol{m}} \varphi_\ell \varphi_{\boldsymbol{m}} = -i \sum_{\ell} \ell_1 c_\ell \varphi_\ell - \frac{1}{Re} \sum_{\ell} (\ell_1^2 + \ell_2^2) a_\ell \varphi_\ell \tag{41}$$

**Theorem 1.1:** Let $u_0$, $F_B$ and $t_f > 0$ be given such that

$$\boldsymbol{u}_0(\boldsymbol{x}) \in H(\Omega), \quad \text{and} \quad \boldsymbol{F}_B(\boldsymbol{x}, t) \in L^2(0, t_f; H) \tag{42}$$

Then $\exists a$ unique solution $u \in (u_1, u_2)^T$

$$u_i, \frac{\partial u_i}{\partial x_j} \in L^2(\Omega \times (0, t_f)), \quad i, j = 1, 2 \tag{43}$$

and $u$ is continuous from $[\theta, t_f]$ into $H$. Moreover, the following energy equation holds for $t \in [\theta, t_f]$:

$$\frac{1}{2} \frac{d}{dt} \|u(t)\|_{L^2}^2 + \nu \|u(t)\|_{H^1}^2 = \langle \boldsymbol{F}_B(t), \boldsymbol{u}(t) \rangle \tag{44}$$

**Theorem 1.2:** Suppose $u_0$, $F_B$ are given and are such that:

$$\boldsymbol{u}_0(\boldsymbol{x}) \in V(\Omega), \quad \text{and} \quad \boldsymbol{F}_B(\boldsymbol{x}, t) \in L^2(0, t_f; H) \tag{45}$$

for $t_f > 0$. Then $\exists$ a unique (strong) solution $u = (u_1, u_2)^T$ with

$$u_i, \frac{\partial u_i}{\partial t}, \frac{\partial u_i}{\partial x_j}, \frac{\partial^2 u_i}{\partial x_j \partial x_k} \in L^2(\Omega \times (0, t_f)), \quad i, j, k = 1, 2 \tag{46}$$

and $u$ is continuous from $[0, t_f]$ into $V$.

## 7.2 SIMPLE ALGORITHM

SIMPLE algorithm is a finite-volume based scheme, which uses a staggered grid which is derived using terms of discrete equations rather than from discretization of a continuous function.

The flow variables can be expressed at any given time step as defined below:

$$u = u^* + u', \quad v = v^* + v' \quad \text{and} \quad p = p^* + p' \tag{47}$$

where $*$ denotes an initial estimate, and $'$ represents a correction term. This can be substituted in the Navier Stokes Equation to produce the equation below:

$$(u^* + u')_t + \left((u^* + u')^2\right)_x + ((v^* + v')(u^* + u'))_y = -(p^* + p')_x + \nu \Delta (u^* + u') \tag{48}$$

We ignore effects of the corrections on advective and diffusive terms thus giving the set of equations below in terms of both x and y momentum:

$$u_t^* + (u^n u^*)_x + (v^n u^*)_y = -p_x^n + \nu \Delta u^* \tag{49}$$

$$u_t' = -p_x' \tag{50}$$

$$v_t^* + (u^n v^*)_x + (v^n v^*)_y = -p_y^n + \nu \Delta v^* \tag{51}$$

$$v_t' = -p_y' \tag{52}$$

The differentiation with respect to time of Pressure Poisson Equation is defined in the equation below:

$$D' = u_x' + v_y' \tag{53}$$

$$\frac{\partial D'}{\partial t} = \frac{\partial}{\partial t}\left(u_x' + v_y'\right) = u_{tx}' + v_{ty}' \tag{54}$$

Using a simple forward-difference approximation, we have:

$$\frac{\partial D'}{\partial t} = \frac{D'^{n+1} - D'^n}{\Delta t} \tag{55}$$

Using the set of Navier-Stokes equations, this can be written as:

$$\frac{\partial D'}{\partial t} = -p_{xx}' - p_{yy}' \tag{56}$$

Next, a PPE for the correction pressure can be obtained as:

$$\Delta p' = \frac{D^*}{\Delta t} \tag{57}$$

The boundary conditions to be employed with this PPE are defined using equations below:

$$p' = 0 \tag{58}$$

$$\frac{\partial p'}{\partial n} = 0 \tag{59}$$

Once $p^{'}$ has been calculated, we can then update the pressure and velocity components using the set of equations below:

$$p^{n+1} = p^n + p'$$ (60)

$$u^{n+1} = u^* - \frac{\partial p'}{\partial x}\Delta t \equiv u^* + u'$$ (61)

$$v^{n+1} = v^* - \frac{\partial p'}{\partial y}\Delta t \equiv v^* + v'$$ (62)

