# OpenReview forum: "AA-PINN: ATTENTION AUGMENTED PHYSICS INFORMED NEURAL NETWORKS"
_ICLR.cc/2022/Conference — ICLR 2022 Submitted_

### Official Review · Reviewer_Ck1m · 2021-10-25

**Correctness:** 3
**Technical Novelty And Significance:** 3
**Empirical Novelty And Significance:** 3
**Recommendation:** 3
**Confidence:** 3

**Main Review:**

This paper is not well organized. In some sections e.g. BACKGROUND, there is a whole page describing the MOMENTUM EQUATIONS and PRESSURE POISSON EQUATION, however later in the AA-PINN experiment, these equations are never used. Also in the Results section, the authors just display the outcomes from the AA-PINN without any discussion about why AA-PINN works. The ablation study of the channel and spatial attention module on the performance looks interesting, but again there is no comment from the authors at all.

I would not recommend this paper to be published in the current form.

My addition comments are as follows:

1.	On page 4, the BURGER’S EQUATION Eq. (18) doesn’t show f(t, x) at all. Later in the loss function Eq. (22) on page 5, f(t, x) is used. This will cause confusion.
2.	The definition of t_u_i is given right after Eq. (18), but it is not used in that part. Instead, it is used later in Eq. (22). This will confuse reader as well.
3.	The terms of t_u_i, x_u_i, u_i, N_u, etc. should be explained respectively instead of as a whole.
4.	On page 4, Figure 1 contains two operations 1) reshape with x, 2) reshape with +. What are they about, respectively?
5.	On page 8, Figure 6 has three rows of plots, what are the last two rows about, respectively?
6.	On page 7, Table 1 shows the performance comparison of the proposed network with previous state of the art. What equation are those networks solving? Where is the data for those neural networks, e.g., TF-net, from?


**Summary Of The Paper:**

The manuscript entitled “AA-PINN: ATTENTION AUGMENTED PHYSICS INFORMED
NEURAL NETWORKS” use a combination of channel and spatial attention module in addition to the normal Physics Informed Neural Networks. Using evaluation metrics like RMSE, divergence and thermal kinetic energy, AA-PINN network outperforms previous PINNs for modelling Navier Stokes and Burgers Equation.


**Summary Of The Review:**

This paper proposed attention augmented physics informed neural networks (AA-PINN). It uses a combination of channel and spatial attention module in addition to the normal Physics Informed Neural Networks. This paper would be an interesting paper however it is poorly organized in the current version and also has a lot of flaws as I pointed out above earlier.

In summary, I would not recommend its publication in ICLR until all my comments are addressed.

---

### Official Review · Reviewer_oeTZ · 2021-10-29

**Correctness:** 2
**Technical Novelty And Significance:** 2
**Empirical Novelty And Significance:** 1
**Recommendation:** 3
**Confidence:** 4

**Main Review:**

PINNs are a vibrant research area that combine ideas from deep learning and numerical methods and scientific computing. Advancing this area is of interest for the scientific machine learning community and work that advances the state-of-the-art can have significant impact. Designing new architectures that are better suited for certain scientific problems is a plausible research direction since (a) standard PINNs rely on very simple fully-connected architectures and (b) architectures that are designed for CV and NLP problems might not be optimal for scientific problems.  Hence, the research question that the authors address is of interest.

However, the paper has several weaknesses and there are several open questions after reading the paper.

1. Method:

* The proposed spatial and channel attention models are insufficiently discussed. It is not clear to follow the logic and what the motivation for the design of these modules is, in the specific context. The concept of attention is not explained.

* It would be helpful to have more details about the 1x1 conv layers to understand whether the dimensions are increased or reduced. I assume that the authors use zero-padding and stride of 1 (details are not provided). Feeding a given tensor x of shape (B, K, H, W) into the proposed spatial attention model that applies twice a 1x1 convolution layer with F_1 and F_2 filters should return a new tensor of shape (B, F_2, H, W), i.e., the spatial attention model is changing the filter dimension from K to F_1 to F_2. Since this output is added to the input tensor x, F_2=K. But, details about F_1 are not provided and it is not unclear whether the number of channels are increased or decreased here, i.e., if this module is introducing some bottleneck.

* It is not clear what is meant by `reshape' and what the addition operation is doing in Figure 1 and 2.

* The ablation study does indicate that there is some effect of using both modules together, but the discussion of the ablation study is insufficient. On which dataset has the study been performed? Why does the model with both modules has less parameters? Does this mean the modules essential reduce the dimensions? Also a baseline model that does not use any additional module is missing.

* Why is the new loss function more robust? How do you tune the many new parameters that are introduced? A detailed ablation study is missing.


2. Experiments

* Following the reasoning behind the experiments and understanding the content in the tables is difficult. First, it is not clear where all the other models are coming from. For example, has a GAN been trained on the specific dataset by someone else, or by the authors? In the former case, some references are needed; in the latter case, more information about the setup is needed.

* Next, I don't know what the authors mean by `accumulated errors'. The author introduce other metrics for evaluation, but I cannot find any table that shows, for instance, the RMSE.

* In Figure 6, what is the difference between the second and third row?


3. General comments

* It would be helpful to clearly state and discuss the model / PINN framework that the authors consider.

* The related work jumps between general deep neural networks for scientific applications and PINNs. It is confusing to follow the related works.

* What do the authors mean by `running at good enough speed'?

* The background section can be moved to the appendix. It is not relevant to understand other parts of the paper.

* The additional provided materials seem not relevant for the general discussion.

* I strongly suggest that the authors expand the discussion and provide additional details to clarify the working mechanisms of both the spatial and channel module.

**Summary Of The Paper:**

The authors propose a new architecture and loss function for training physics-informed neural networks (PINNs) on fluid flow problems. The idea is centered around augmenting a basic residual network with two additional attention blocks that are placed before and after residual blocks. These two new blocks aim to introduce channel and spatial attention into the model. In addition, they propose a new loss function that is tailored to solving PDEs such as Navier Stokes and Burgers Equation.

**Summary Of The Review:**

The authors introduce a new architecture for PINNs, which can be seen to be novel up to a certain extend. However, modules of similar flavor have been proposed in previous works and simply proposing a new architecture that augments an existing architecture with additional modules or layers is not sufficient. The experiments need to clearly study and demonstrate the advantages on a range of problems, in particular if no theoretical justifications are provided. The provided experiments do not live up to the standards that I would expect from an empirical paper. Further, the presentation and discussion of the ideas is insufficient and many details are missing. Also, no research code is provided to reproduce or better understand the proposed architecture. The overall quality of this paper is below the acceptance threshold for ICLR.

---

### Official Review · Reviewer_kPHW · 2021-11-02

**Correctness:** 3
**Technical Novelty And Significance:** 2
**Empirical Novelty And Significance:** 2
**Recommendation:** 3
**Confidence:** 4

**Main Review:**

I appreciate the idea of combining attention mechanisms with physics-informed neural networks, but I expect more than merely an addition. Also, there are a few weaknesses I have to point out:

1. The author did not provide any explanation on the intuition behind refactorizing the channel and spatial dimensions. Is it for memory efficiency? How does it compare to non-local neural networks?
2. Physical process usually comes with tons of hyperparameters, which is understandable because of the complexity. However, a lot of the hyperparameters are left unexplained or unexperimented. For example, how were $\alpha$ and $\beta$ selected? What does the overall network architecture look like? Does the number of attention modules stacked matter to the result?
3. One key question to physical simulation with neural networks is generalizability. It will be very helpful to demonstrate how this method generalizes to unseen boundary conditions and initial fields.
4. Attention mechanisms intrigue me by not only a superior performance on multiple tasks but also the explainability. It will be interesting to see the visualization of them so that it might shed some light on what and how the networks learn physics.

In addition to these technical unclearnesses, my main concern is the novelty of mixing multiple popular topics. I expect more solid theories or more exciting demonstrations.

**Summary Of The Paper:**

This manuscript introduces the attention module into the framework of physics-informed neural networks. The contributions are: (1) proposing a network architecture that marries the popular attention mechanism to physics-informed neural networks. (2) separating the large attention module by operating on the channel and spatial dimensions individually. The authors empirically studied the efficacy of this architecture on PDEs including Navier-Stokes fluid simulation and Burgers equation.

**Summary Of The Review:**

This manuscript talks about two popular topics: attention mechanisms and physics-informed neural networks. I believe something significant will happen in this particular overlap but not in a "1+1=2" form. I expect more explanatory theories or experiments and domain-specific modifications.

---

### Official Review · Reviewer_qXxf · 2021-11-02

**Correctness:** 3
**Technical Novelty And Significance:** 2
**Empirical Novelty And Significance:** 2
**Recommendation:** 3
**Confidence:** 3

**Main Review:**



The presentation is very poor, experimental details are missing, and it is not clear
against which models the authors are benchmarking. They report increased performance
according to a metric they don't even define.

- Missing references in the introduction

- Open problems in the field are not discussed and
motivation for the proposed model is missing.

- Poor English throughout the paper and generally a poor presentation.
The first sentence
"Computational Fluid Dynamics (CFD) has become the core technology behind almost every fluid
simulation." is almost tautologic. Many ill-composed sentences such as
"A thorough study of data-driven methods using machine learning approaches for
modeling the turbulence was studied by ...".

- tilde and bar variables not defined in Eq. 4

- No outline of the paper in the introduction. Not clear what is the purpose of some sections like 3.2 and 3.3.

- The author should care to explain what is the output of the network.

- Not clear what models the authors are benchmarking against, as
they don't provide details and references.
 Possibly they should apply their attention augmentation to  State of the art
 architectures such as the ones analyzed in the NSFNets paper.

- The authors say they evaluate on a few different metrics, but all I can see reported is
"accumulated errors", which I have no idea what it is since it has not been defined.

- A figure is not a detailed explanation of a NN block but only a visual help, please provide definitions in formulas.

**Summary Of The Paper:**

The authors consider Navier-Stokes equation in 2D and the 1D Burger's equation.
Using a PINN approach to model the solution with neural networks, they propose an architecture that alternates spatial and channel attention blocks to convolutional blocks.


**Summary Of The Review:**

The proposed model may have some selling points to it, but the presentation
is so poor and so many details are missing that it is impossible to tell.
This manuscript is far from being up to ICLR standards.

---

### Decision · Program_Chairs · 2022-01-20

**Decision:**

Reject

**Comment:**

All four reviewers agree that the paper should be rejected in its current form, but make numerous suggestions for improving it. The main points of concern were the motivation of the proposed method, novelty and the quality of the presentation of the work. The authors did not provide a response. The AC agrees with the reviewers and recommends rejecting the paper.